# Fluid Interfaces as Models for the Study of Lipid-Based Films with Biophysical Relevance

Pablo G. Argudo [1], Armando Maestro [2,3] and Eduardo Guzmán [4,5,*]

1 Max Planck Institute for Polymer Research, Ackermannweg 10, 55128 Mainz, Germany; argudop@mpip-mainz.mpg.de
2 Centro de Física de Materiales (CSIC, UPV/EHU)-Materials Physics Center MPC, Paseo Manuel de Lardizabal 5, 20018 San Sebastián, Spain; armando.maestro@ehu.eus
3 IKERBASQUE—Basque Foundation for Science, Plaza Euskadi 5, 48009 Bilbao, Spain
4 Departamento de Química Física, Universidad Complutense de Madrid, Ciudad Universitaria s/n, 28040 Madrid, Spain
5 Instituto Pluridisciplinar, Universidad Complutense de Madrid, Paseo Juan XXIII 1, 28040 Madrid, Spain
* Correspondence: eduardogs@quim.ucm.es; Tel.: +34-91-3944107

**Abstract:** This comprehensive review aims to provide a deep insight into the fascinating field of biophysics in living organisms through the study of model fluid interfaces that mimic specific lipid-based structures with biophysical relevance. By delving into the study of these physiological fluid interfaces, we can unravel crucial aspects of their behavior, lateral organization, and functions. Through specific examples, we will uncover the intricate dynamics at play and shed light on potential pathogenic conditions that may result from alterations in these interfaces. A deeper understanding of these aspects can lead to the emergence of novel technologies and medical advances, potentially leading to the development of innovative treatments and diagnostic tools.

**Keywords:** biointerfaces; fluid interfaces; Langmuir films; lipids; monolayers; proteins

## 1. Introduction

Interfacial aspects are widespread in different phenomena affecting the physiology of living organisms, e.g., lung surfactant performance, tear film wettability, lipid digestion or biomembrane organization [1–4], as well as in various processes of biomedical relevance [5,6]. This has stimulated an intense research activity aimed at exploiting the tools provided by colloid and interface science to gain a better understanding of various processes and/or phenomena of biophysical relevance. It is true that colloid and interface science generally provides very simple models, but these can help to elucidate some of the most fundamental physicochemical aspects underlying the physiological response of various biologically relevant systems [1,7]. For this purpose, the use of lipid (or lipid-protein mixtures) layers deposited at fluid interfaces, and, in particular, at water/vapor interfaces, or their deposition on solid surfaces forming Langmuir-Blodgett (LB) or Langmuir-Schafer (LS) films, are simple models of paramount importance. In addition, these models may be used to understand how specific chemicals (e.g., contaminants, pharmaceuticals, etc.) modify the structure–function relationship of biomimetic systems [5,8–15]. However, these model systems are an oversimplification of the complex biophysical framework running in vivo and allow only certain aspects to be mimicked. This is partly because the number of components to be incorporated in model fluid layers is reduced in relation to the high chemical complexity present in biological systems. In fact, model fluid layers are minimal systems. In addition, model systems do not allow to consider specific boundary conditions, e.g., transport phenomena, mechanical aspects, or geometric and morphological constraints, that occur in vivo [2,3]. However, most of the biological systems that are commonly modeled by using monolayers are not truly a single leaflet but appear as

multilayered structures or thin films at the interface combined with 3D structures in the bulk [2]. Therefore, extrapolation of results from model systems to the in vivo situation is not always straightforward.

Overall, this review aims to provide a coherent and insightful overview of the significance of model fluid interfaces in biophysics and biology. By examining two specific examples of lipid-based films and exploring the consequences of changes in their organization, we hope to make a significant contribution to the broader understanding of biological phenomena. Furthermore, we believe that the knowledge gained from this review has the potential to drive transformative developments in biotechnology, further enhancing our ability to improve human health and well-being. It is important to emphasize that living organisms contain a variety of examples of lipid-based layers (Table 1 lists six significant examples). Nonetheless, within the context of this review, we have opted to concentrate our attention on the lipid layer within the tear film and the surfactant films found in the lungs. This choice is not arbitrary, but it is based on the similarities in composition and the role of mechanical properties in their normal physiological function [16]. As a result, we believe that these two systems serve as excellent examples to elucidate the behavior and potential pathogenic situations that may result from changes in the organization and dynamics of such model systems.

**Table 1.** Overview of common lipid-based layers in living organisms. Adapted with permission from Ref. [16]. 2013, Elsevier.

| Lipid-Based Layer | Re-Spreading | Major Components |
| --- | --- | --- |
| Tear Film lipid layer | Yes | Wax esters, Cholesteryl esters, Polar lipids, Surfactant proteins |
| Lung surfactant | Yes | Phospholipids, Surfactant proteins |
| Human skin lipid | No | Ceramides, Cholesterol, Fatty acids |
| Skin lipid secretions (tree frog) | No | Wax esters, Triglycerides |
| Plant lipid layer (cuticle) | No | Wax esters, Triglycerides |
| Arthropod lipid layer | No | Hydrocarbons |

## 2. Applications of Lipid Films and Fluid Interfaces

Lipids play a central role in the composition of plasma membranes. These membranes consist of a polar hydrophilic headgroup and hydrophobic tail(s). The lipid bilayer present in plasma membranes serves to create a barrier between intracellular components and the external surroundings. It also serves as a protective enclosure for membrane proteins while selectively allowing the passage of specific ions or molecules. The remarkable abilities of plasma membranes to sense, detect, and facilitate the transport of particular substances have intrigued the scientific community, driving active investigation into the functional aspects of lipid bilayers [17]. In aqueous solutions, lipid films enclosed within structures such as lipid vesicles or liposomes have gained significant use as "carriers" for delivering drugs and genetic material. This is due to their unique capacity to incorporate both hydrophilic and lipophilic molecules within separate compartments [18]. Additionally, lipid bilayers that are supported on solid surfaces, referred to as supported lipid bilayers, serve as valuable models for plasma membranes. These models have been extensively utilized to explore fundamental cellular processes, notably interactions between lipids and proteins [19].

While flat lipid bilayers constitute the predominant configuration of lipids in the natural context, it is important to note that diverse lipid types have the inherent ability to self-organize into a range of more intricate structures [20]. These encompass morphologies that become progressively more intricate, including micellar formations, 2D

arrangements, and 3D phases [21]. The capacity of lipids to form a variety of structures, a phenomenon referred to as lipid polymorphism, has garnered significant interest due to the distinct properties it imparts in a phase-dependent manner. Most research exploiting lipid polymorphism has concentrated on specific lipid systems. However, recent investigations indicate that this phenomenon can be extended to lipid assemblies confined to surfaces/interface [22]. The exploration of lipid films and surface-mediated phase transitions is still in its early stages, primarily because studies involving lipid films have mainly focused on mimicking cellular membranes using a single layer model [23].

It is important to highlight the growing interest in developing larger-scale implantable drug delivery devices [24], high throughput biosensing systems [25], and substrate-mediated drug/gene delivery [26]. Progress in biotechnology, including techniques for surface patterning [27], microfluidics [28], and biodegradable organic electronics [29], has greatly enriched research in these applications. Consequently, there is a rising demand for biointerface membranes, acting as matrices or scaffolds that offer precise control over the spatiotemporal release of payloads. This trend underscores the swift emergence of the need for the study and characterization of lipid films [30].

The combination of lipid films with fluid interfaces is of crucial importance because both types of systems are ubiquitous within the human body, playing integral roles in various physiological processes. Despite the diverse array of functions they serve, the underlying structures and phenomena dictating their physiological efficacy remain fundamentally consistent. Moreover, the study of lipid films at fluid interfaces can help on the development of different applications of lipid-based structures [1]. For instance, this can be exploited for development specific applications of lipid-based materials. In fact, understanding fluid interfaces aids in optimizing the structure and stability of lipid-based drug carriers like liposomes. Insights into the behavior of lipid films at interfaces can guide the design of liposomal formulations for improved drug release profiles and targeted delivery. In the same direction, it operates the efforts of the cosmetic industry for obtaining lipid-based nanostructures with optimal surface properties, therefore obtaining products with improved stability and controlled release of active ingredients, leading to enhanced cosmetic benefits. However, fluid interface studies aid in tailoring lipid-based films for specific food applications. Knowledge of interfacial properties can aid in creating films that effectively protect food from external factors while maintaining desirable texture and appearance.

It should be noted that the method of fabrication using liquid/liquid interface as a template provides a remarkable control over the mesostructure of the resulting product. This approach circumvents a prevalent issue encountered in numerous film synthesis methods, where the reaction takes place at a liquid/solid interface [31].

## 3. Surface Science Methods for Studying Biophysical Relevant Systems

Colloidal and interfacial approaches provide valuable insights into the physicochemical behavior of various systems of biological interest, overcoming the limitations of invasive methods [5,10,32]. The use of Langmuir film balances [10,33], although not providing direct biophysical information, helps to understand changes in surface pressure and molecular area plot, the so-called interfacial isotherm [34–39]. These balances also allow the study of rheological responses during compression–expansion cycles [40,41]. However, the frequencies and deformation amplitudes probed differ from those naturally occurring in biological systems and provide only semi-quantitative information about the true behavior of biophysically relevant systems. Langmuir film balances can also be used to transfer interfacial films onto solid substrates, allowing ex situ studies using microscopy techniques such as atomic force microscopy (AFM) [42–45]. See Figure 1 for sketches of the Langmuir film balance and film deposition on solid surfaces.

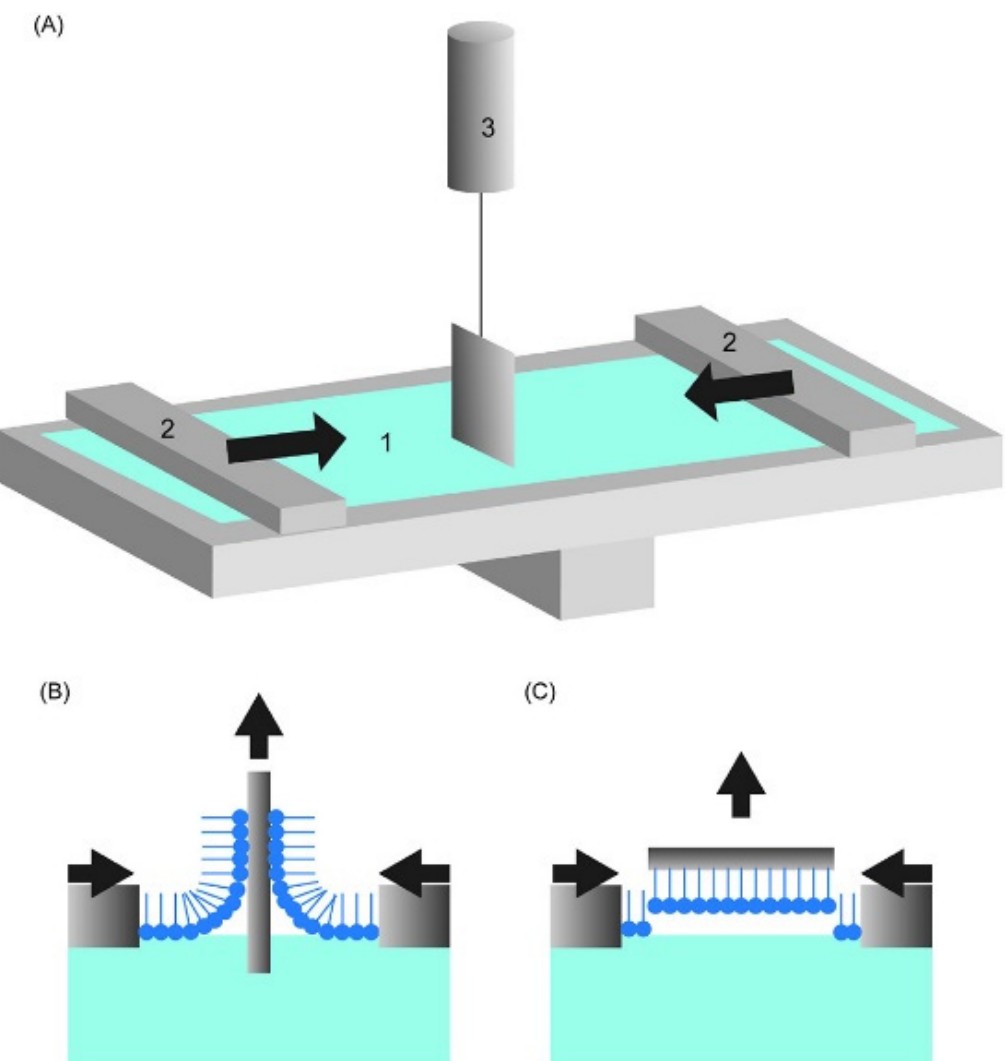

**Figure 1.** (**A**) Schematic of a Langmuir film balance setup with a Wilhelmy plate contact probe and two symmetrical barriers. (**B**) Vertical transfer of a film from a fluid interface to a solid substrate using the Langmuir–Blodgett method. (**C**) Horizontal transfer of a film from a fluid interface to a solid substrate using the Langmuir–Schaefer method. Reprinted with permission from Ref. [45]. 2020, Elsevier.

The highly dynamic nature of most systems with biophysical relevance requires a deeper understanding of the mechanical properties of biologically relevant films [46,47]. While Langmuir film balances have limitations in providing this information (for further details, see Ravera et al. [48]), techniques such as pulsating bubble surfactometers or captive bubble tensiometers [49–53] help to assess the film mechanics under realistic physiological simulations. These methods allow for the assessment of film adsorption, remodeling, surface activity, and stability [54,55]. Oscillating bubbles and drops can also be used to study biologically relevant systems [56–58]. These approaches provide insight into properties such as minimum interfacial tension, hysteresis loop area, and dilational viscoelasticity, which help to understand how pathological states modify the kinetics and mechanics of biologically relevant films [59–62]. The constrained drop surfactometer uses a single sessile droplet positioned on a knife-edge platform that effectively confines the film without the risk of leakage. This setup allows real-time assessment of volume, surface area, and stress through closed-loop axisymmetric drop analysis. It also allows for precise harmonic deformations of the interface with predefined amplitudes and frequencies, which are valuable in determining the surface Young's modulus [63]. In addition, interfacial shear

rheology can also provide important relationships between the mechanics of biologically relevant interfacial films and their structure and organization [64,65].

Assessing interfacial properties offers valuable insights into how pathological conditions influence the dynamic behavior of biologically significant layers [66]. Comprehending the connection between alterations in the packing of biologically significant film interfaces and the deterioration of mechanical characteristics is of utmost importance [67,68]. Techniques such as Brewster Angle Microscopy (BAM), Atomic Force Microscopy (AFM), Surface Force Apparatus (SFA), Ellipsometry, Infrared Reflection Absorption Spectroscopy (IRRAS), or Epifluorescence Microscopy, and their combinations with tensiometric techniques (mainly Langmuir film balances), are powerful to study the changes in biological relevant film organization at micrometric and submicrometric scales [69–72]. Advanced tools such as neutron reflectivity or synchrotron grazing angle X-ray diffraction can also help to evaluate the organization of biologically relevant interfacial films [39,73–84].

## 4. The Tear Film and the Dry Eye Disease

The tear film is a fluid layer that covers the corneal surface and protects the eyes from the environment. It has a heterogeneous structure that can be divided into three layers [85]. The innermost layer (mucus layer) is a gel-like mucin layer (2.5–5 μm thick), composed mainly of glycosylated proteins produced by epithelial cells. The role of this inner layer is to provide an easily wettable surface that contributes to the easy re-distribution of the tear film between blinks. The intermediate layer, which measures approximately 4 μm in thickness, comprises a diverse mixture of substances that encompass both water-soluble and water-insoluble components. This assortment includes electrolytes, proteins, peptides, small-molecule metabolites, and mucins [86,87]. The intermediate layer is characterized by its smooth surface, enabling light refraction. Furthermore, it serves multiple functions such as facilitating lubrication during blinking and eye movements, preventing ocular surface dehydration, offering protection against environmental pathogens and particles, and aiding in the nourishment of corneal cells [87]. The outermost layer of the tear film, known as the tear film lipid layer (TFLL), is the thinnest component, ranging in thickness from 0.015 μm to 0.160 μm, and is predominantly composed of lipid compounds [88]. The composition, structure, and function of the TFLL are still controversial, but it is generally accepted that its key role is related to the reduction of the interfacial tension of the tear film. In addition, its presence is of a paramount importance in ensuring tear film stability [89]. Figure 2 shows a sketch of the multilayered structure of the tear film.

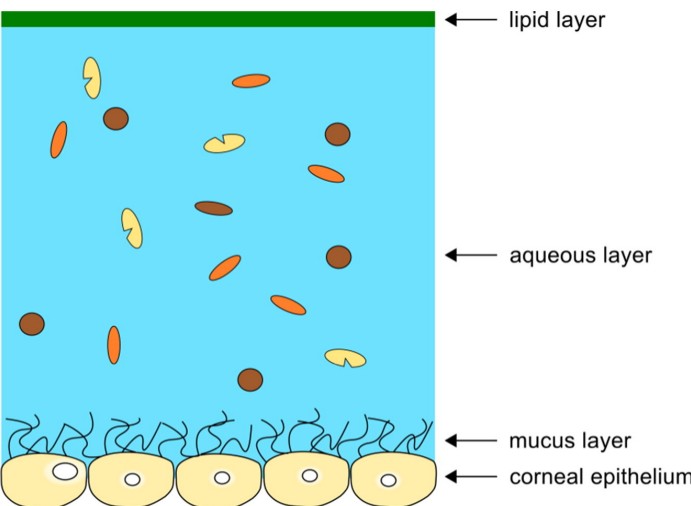

**Figure 2.** Sketch of the multilayered structure of the tear film. Reprinted with permission from Ref. [89]. 2016, Elsevier.

It is important to emphasize that the tear film is a dynamic system influenced by three major processes: tear flow, evaporation, and blinking. While tear flow and evaporation help to maintain a relatively stable state over time, blinking introduces significant irregular perturbations to the tear film. As a result, the tear film is not a static entity, but rather a complex interplay of these processes that constantly adapt to maintain ocular health and clarity [90]. After each blink, the tear film undergoes a crucial process of re-spreading across the corneal surface, with certain aspects not yet fully understood. Several factors, including osmolality, gravitational forces, and evaporation, have been identified as influencing the phenomenon of tear film spreading. Understanding these mechanisms is essential to understanding the complex dynamics that contribute to maintaining tear film stability and ocular health [91].

### 4.1. Tear Film Lipid Layer (TFLL)

As mentioned previously, the tear film lipid layer (TFLL) occupies the outermost position within the tear film, located at the interface between the eye and the surrounding air. Its primary function is to reduce the interfacial tension of the tear film, thereby providing stability on the corneal surface [92]. In addition, the TFLL plays a crucial role in smoothing the surface of the tear film. This smoothing effect helps to slow down water evaporation, which helps to maintain the stability of the tear film. In addition, the TFLL acts as a barrier that prevents the tear film from overflowing at the eyelid margins, thus ensuring the integrity of the tear film and preventing discomfort or potential irritation [89]. However, the TFLL also plays a vital role in protecting the eyes from pathogens [93]. From a biochemical point of view, the TFLL is not only composed of lipids but also presents intercalated proteins that contribute to modulate the properties and function of the TFLL [94].

It is important to note that the TFLL cannot be considered as a simple monolayer, but it has a multilayered structure with lateral heterogeneity [88]. The widely accepted structural model for the TFLL proposes the existence of two distinct regions [95]. The innermost region, which is in direct contact with the aqueous layer of the tear film, consists of a densely packed layer of polar amphiphilic lipids 2-to-9-nm thick. In contrast, the outermost region, located at the eye-air interface, consists of a thicker layer of 33 to 40 nm, characterized by a more disordered structure and composed of non-polar hydrophobic lipids [85,96]. Figure 3 displays a sketch of the TFLL, showing the two characteristic regions.

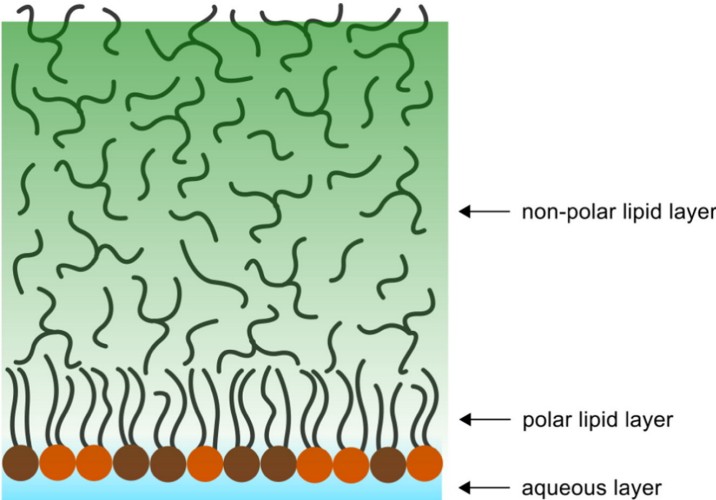

**Figure 3.** Sketch of the structure of the lipid layer of the tear film. Reprinted with permission from Ref. [89]. 2016, Elsevier.

The decrease in tear film thickness after blinking is often attributed to the evaporation of water from the underlying aqueous subphase [97,98]. However, post-blink tear film degradation results in detectable changes in the diffraction patterns, indicating changes

in the structure of the tear film lipid layer (TFLL) [91]. It is reasonable to assume that the dynamic processes primarily affecting the TFLL are associated with eyelid movements. These movements are likely to involve both lateral compression and decompression of the lipids, as well as direct interactions with the lipid layer from above. However, the exact details of how the eyelids affect the TFLL are not fully understood. Some theories suggest that the eyelids either slide over the lipid film, acting as a lubricant [93], or compress the film, requiring it to be completely redistributed as the eyelids open [89]. The physiological blink is thought to involve a mixture of these influences, resulting in a remarkable reconfiguration of the lipid layer. Unfortunately, the intricacies of these mechanisms are not fully understood, largely due to the inherent complexity of the experimental investigation. Some insight can be gained from Langmuir balance experiments involving lateral compression–expansion cycles, while molecular dynamics (MD) simulations also provide a means to gain a partial understanding of TFLL dynamics [99].

One of the most intriguing aspects of TFLL research is its association with dry eye syndrome (DES). In the existing literature, the loss of the anti-evaporation properties of the lipid layer is often associated with this ocular condition. Extensive lipidomic studies have been performed to compare dry eye patients with the general population in order to establish a detailed relationship between dry eye syndrome (DES) and tear film lipid layer (TFLL) properties. However, current experimental methods have limitations and, as a result, a clear relationship between TFLL composition and the occurrence of DES remains elusive [100].

### 4.1.1. Tear Film Lipid Layer Composition

The composition of the TFLL is qualitatively similar to that of the other biologically relevant layers, e.g., lung surfactant (discussed below), containing polar lipids and surface-active proteins [101,102]. However, the role of the latter is unclear, although it is commonly accepted that they play an important role in the reorganization of the TFLL during the blink cycle [16].

For a long time, most of the lipids contained in the TFLL were thought to be secreted by the meibomian glands [103]. However, recent lipidomic studies have challenged this belief, as the composition of the whole tear film fluid differs significantly from that of meibomian lipids [104]. In particular, the phospholipids in total tears range from 5–13 mol%, while in the meibomian gland secretion, they are up to 0.1 mol%. The origin of the phospholipids in the tear film is still unclear [100,105,106]. They play a key role in the TFLL as the major component of the polar sublayer, stabilizing both the TFLL and the aqueous tear subphase [105,107]. However, there is an ongoing debate regarding the exact contributions of meibomian and non-meibomian polar lipids to the tear film composition [105].

The non-polar lipids produced by the meibomian glands comprise approximately 30–45 mol% of cholesterol esters (CE) with long acyl chains (C22:1–C34:1), about 30–50 mol% of wax esters (WE) composed of C18:1 fatty acid chains in conjunction with C18–C30 alcohol chains, and up to 4 mol% (O-acyl)-u-hydroxy fatty acids. Non-meibomian polar lipids, which make up approximately 13 mol% of TFLL, consist mainly of glycerophospholipids, lysophospholipids, and sphingomyelins [100,108]. The most abundant lipid classes in the tear film are phosphatidylcholines (PCs) (>60 mol%) and phosphatidylethanolamines (PEs) (~15%), together with small amounts (<5%) of ceramides (Cer) and sphingomyelins (SM) [108].

### 4.1.2. Tear Film Lipid Layer Studies Based on Interfacial Layers

The ideal TFLL mimicking layer must meet specific requirements: (i) High evaporation resistance to prevent water loss and hyperosmolarity; (ii) Excellent respreading after each compression–expansion cycle; (iii) Sufficient fluidity to avoid interfering with meibomian gland secretion; and (iv) A gel-like, incompressible nature to withstand disruptive forces. However, certain properties of the lipid layer tend to be incompatible, such as the conflict between gel-like incompressibility and fluidity, and high evaporation resistance leading to

poor spreadability. As a result, the composition and structure of the lipid layer requires a compromise and trade-off between these conflicting properties [16].

In a study by Olżyńska et al. [109], they investigated a minimal model of TFLL consisting of a binary mixture of polar and non-polar lipids (1-palmitoyl-2-oleyl-sn-glycero-3-phosphocholine (POPC) and glyceryl trioleate (TO)). The results showed a significant structural reorganization of the model film influenced by lateral pressure and the ratio of polar to non-polar lipids. At low lateral pressure, the film appeared single-molecule thick and uniform, while compression led to a transformation into a multi-layer structure with polar–non-polar lipid assemblies. According to this framework, it is proposed that the in vivo tear film lipid layer (TFLL) is likely to have a bilayer structure with both polar and non-polar components. In addition, it is likely to contain multiple mixed lipid aggregates that are formed as the film restructures. Although the model is simple in design, these findings have implications for advancing our understanding of TFLL physiology and conditions associated with lipid imbalances within the tear film.

Xu et al. [110] conducted an investigation of the relationships between composition and function in an artificial tear film lipid layer (TFLL) under conditions that mimic physiological conditions. They created this artificial TFLL using a blend of 40% behenyl oleate and 40% cholesteryl oleate, representing the most common wax and cholesteryl esters found in natural TFLL. In addition, it contained 15% phosphatidylcholine and 5% palmitic acid-9-hydroxystearic acid (PAHSA), representing the two major classes of polar lipids found in natural TFLL. The results of their study indicated that phospholipids were primarily responsible for reducing interfacial tension, while PAHSA played a key role in optimizing the rheological properties of TFLL.

Keramatnejad and De Wolf [111] conducted a comprehensive investigation to explore the biophysical properties of model membranes that mimic the tear film lipid layer (TFLL) by systematically varying their composition. They examined three mixed-lipid model membranes and their individual lipid components, including cholesteryl oleate (CO), glyceryl trioleate (GT), L-α-phosphatidylcholine (egg PC), and a mixture of free fatty acids. In this study, the complexity of the models was progressively increased to allow the role of each lipid component to be assessed. They used various techniques, including Langmuir balance, Brewster angle microscopy (BAM), and profile analysis tensiometer (PAT), were used to study the surface activity, compression–expansion cycles, morphology, and rheological behavior of the model membranes. The inclusion of CO-induced multilayering and the GT phase transition resulted in reversible collapse. The addition of polar PC initially resulted in a more cohesive film. Importantly, these individual behaviors were maintained in the mixed films, suggesting a potential role for each physiological component of the TFLL. Although binary mixtures offered insights into the roles of polar phospholipids and cholesteryl esters in TFLL, they did not fully replicate the properties of the natural meibum lipid film. The multilayering of cholesteryl esters in the mixture yielded less reproducible rheological measurements, increased hysteresis in cycling experiments, and non-ideal mixing behavior. Additionally, the addition of as little as 10% phospholipid to cholesteryl oleate initiated early multilayering behavior, possibly due to phospholipid chain packing. Despite the physiological cholesteryl ester content being less than 90%, comparing binary and ternary blends provided insights into the roles of cholesteryl esters, triglycerides, and phospholipids in TFLL. The inclusion of saturated acyl chain free fatty acids (FFAs) raised film viscosity beyond physiological TFLL levels, reduced spreadability, and introduced increased non-ideality in the films. Paananen et al. [112] studied the organization of TFLL using mixed films of polar O-acyl-ω-hydroxy fatty acids (OAHFA) or phospholipids and nonpolar cholesteryl esters (CE) as a model. Their study used Brewster angle and fluorescence microscopy in a Langmuir trough system to investigate film organization and evaporation resistance.

Surprisingly, the results showed that phospholipids and OAHFAs induced the formation of a stable multilamellar CE film even at low surface pressures with low surface concentrations of polar lipids. This multilayer formation was driven by the interdigitation

of acyl chains between the multilayer and the polar monolayer, resulting in the formation of a single multilayer lamella. However, at high surface pressures, the loss of interlayer interdigitation destabilized the multilayer structure. This finding has significant implications for the tear film, which typically has a relatively high surface pressure, primarily due to the presence of polar lipids, although surfactant proteins may also contribute. The study suggests that a stable, uniform, non-polar lipid layer is unlikely to form on the surface of the tear film. Instead, specific regions are likely to contain thick non-polar aggregates as a coating. Furthermore, the CE multilayer did not show a significant improvement in water evaporation resistance compared to a polar lipid monolayer.

Paananen and Cwiklik [113] proposed an in silico model incorporating the major lipid components of TFLL, with wax esters and cholesteryl esters as the primary non-polar lipids and (O-acyl)-ω-hydroxy fatty acids as the primary polar lipids. This model was designed to provide molecular insights into the structure of TFLL, allowing for a direct comparison with experimental results from Langmuir monolayer experiments. The results showed good agreement with experimental data for each lipid class individually. Simulations of mixed systems showed that OAHFA enhanced the spreading of non-polar waxes and cholesteryl esters, resulting in disordered films. Furthermore, the lateral compression of mixed films led to the development of crystalline order in the systems studied. The above results contrast with those reported by Viitaja et al. [114]. They found that the combination of solid-state wax esters with OAHFA can lead to the formation of high evaporation resistance films. This supports the idea that the amphiphilic sublayer present between the aqueous tear film and the lipid layer contributes to the evaporation resistance of the TFLL. The incorporation of non-polar wax esters in this layer may further enhance the evaporation resistance. Cwiklik et al. [115] extended their in silico work on the effect of lipid nature on the lateral organization of the TFLL and found that under typical physiological conditions, long and unsaturated non-polar lipids form a film with a distinct sublayer of polar lipids separating it from the aqueous subphase. However, when the non-polar lipids had shorter and saturated chains, they penetrated into the polar sublayer, altering its properties and the overall structure of the lipid film. These results suggest that the chain length and saturation of non-polar lipids play a role in modifying the structure and properties of the TFLL.

Xu et al. [116] studied a TFLL model based on a mixture containing 40% wax esters, 40% cholesteryl esters, and 20% polar lipids. This mixture was found to reduce the interfacial tension of the water/vapor interface down to 47 mN/m, forming a structure consisting of discrete droplets/aggregates of the non-polar lipids on a polar lipid monolayer with phase separation. Such a structure helps to prevent water evaporation. This is reflected in Figure 4, where the dependence of the evaporation rate and the area per molecule of the lipid at the water/vapor interface are plotted as a function of the surface pressure of the interface ($\Pi = \gamma_0 - \gamma$, where $\gamma_0$ and $\gamma$ are the interfacial tensions for bare the water/vapor interface (about 72 mN/m at 37 °C) and the monolayer, respectively).

Yoshida et al. [117] conducted a study to understand the influence of the endogenous meibum components on the structure and stability of the TFLL. They used Langmuir film balance and Brewster angle microscopy to analyze meibum films from rabbit eyelids and their fractions. Their results showed that meibum films had high stability, with almost identical isotherms during repeated compression–expansion cycles. Brewster angle microscopy revealed the presence of condensed phase network structures composed primarily of wax esters and cholesteryl esters as non-polar components, coexisting with a monolayer phase formed by polar lipids trapped within the networks. This microphase separation arises from the significant difference in intermolecular interactions between the two lipids contained in each phase, with different factors influencing the surface structure during film formation. The networks of non-polar lipids are observed to be located at the edges of the monolayer phase regions. This suggests that the initial step in network structure formation involves the growth of monolayer phase domains of polar lipids, which pushes the non-polar lipids outward. Subsequently, two-dimensional networks are formed by the aggregation of non-polar lipids at the boundaries between the monolayer phase domains,

leading to the confinement of the monolayers. Thus, the condensed phase networks and the confined monolayer regions are formed in an interdependent manner during the microphase separation process in the meibum film. Temperature changes (20 °C and 35 °C) and the presence of salts in the subphase affected the surface pressure and expansion of the films, but the overall stability and morphology of the films remained largely unchanged. Meibum films without phospholipids or with both phospholipids and cholesterol showed marked changes in hysteresis and film morphology, indicating their importance in maintaining film stability against mechanical perturbations. In conclusion, the study by Yoshida et al. [117] highlights the role of specific meibum components in the structural integrity and stability of the TFLL and provides valuable insights into tear film dynamics.

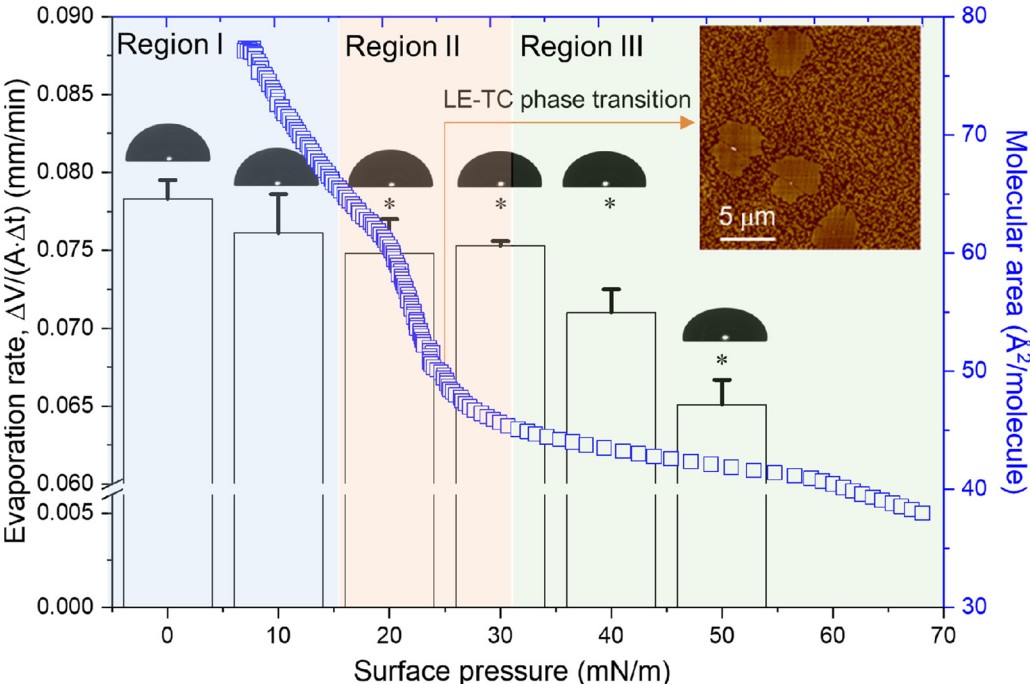

**Figure 4.** The dependence of the evaporation rate and the area per molecule of the lipid at the water/vapor interface are depicted as function of the surface pressure. The inset shows an AFM micrograph depicting the lateral organization of the lipid at the interface. Reprinted from Xu et al. [116].

It is important to emphasize that it can be challenging to apply the knowledge gained from Langmuir films to the actual biophysical conditions within the tear film lipid layer (TFLL) due to the differences in the models commonly used in such studies. From the above discussion, it can be concluded that wax and cholesteryl esters, together with O-acyl-ω-hydroxy fatty acids and their mutual interactions and specific organization, play a crucial role in controlling the properties and physiological function of the TFLL. However, the role of each component in the normal biological function of the TFLL is far from clear. It should be emphasized that the effective functionality of TFLL relies on the adequate presence of each lipid family, characterized by precise chemical structures, arrangement, and proportions. In particular, non-polar lipids are crucial in preventing tear evaporation due to their proximity to the air interface. In addition, specific non-polar lipids, such as triacylglycerols and short-chain wax esters, play a central role in bridging nonpolar and polar layers. However, polar lipids are located at the interface between the aqueous/mucin phase and non-polar lipids and contribute to the stabilization of the TFLL. In addition, phospholipids and O-acyl-ω-hydroxy fatty acids are good surfactants that help maintain a low surface tension, which ensures a low free energy on the ocular surface [118].

### 4.2. The Dry Eye Syndrome

Dry eye syndrome (DES) is a common condition characterized by an imbalance in the tear film. A major factor contributing factor to DES is thought to be alterations in the tear film lipid layer (TFLL), the outermost layer of the tear film, leading to excessive water evaporation and drying of the ocular surface [119]. This results in an unstable tear film, hyperosmolarity, and ocular surface damage. It affects 10–30% of adults and is a significant economic burden, with an estimated annual cost of USD 55 billion in the United States alone [120,121].

Langmuir Films for Understanding the Dry Eye Syndrome

Understanding the relationship between the compromised TFLL structure and DES can aid in prevention planning and improvement of ocular health, and in this context, the use of Langmuir film-based models is a very promising tool.

By integrating atomistic simulations with Brewster angle microscopy and surface potential measurements, Paananen et al. [122] studied the arrangement and evaporation resistance of films containing wax esters, a key component of the TFLL. The results proved that the evaporation resistance of the TFLL is due to crystalline wax ester layers, and that the rate of evaporation depends on both TFLL defects and its coverage of the ocular surface. These results suggest that understanding the non-equilibrium spreading and crystallization of TFLL films may contribute to design strategies for the treatment of dry eye syndrome (DES). Deepening on the role of lipid composition and organization in the TFLL, Bland et al. [119] showed that the key role of the chain length and polar groups in the phase behavior of the TFLL. In particular, the existence of a liquid-expanded-to-solid-phase transition in films containing OAHFA is crucial for the evaporation resistance of TFFL. However, an increase in the concentration of diesters is harmful for the correct physiological performance of the TFLL, contributing to the emergence of DES.

Keramatnejad [123] and De Wolf conducted a study to investigate the effect of external factors, such as ozone levels, on the modification of the structure and properties of the tear film lipid layer (TFLL) and the occurrence of dry eye syndrome (DES). They used up to three different model Langmuir membranes: (i) A binary mixture consisting of cholesteryl oleate (CO) and L-$\alpha$-phosphatidylcholine (egg PC); (ii) A ternary mixture consisting of CO, glyceryl trioleate (GT), and PC; and (iii) A quaternary mixture consisting of CO, GT, a mixture of free fatty acids, including palmitic acid and stearic acid (FFAs), and PC.

Their study showed that ozone exposure resulted in significant changes in the surface properties and behavior of the TFLL model membranes. This was characterized by increased surface pressure and expansion toward higher molecular weight regions in the surface pressure-area isotherms of the membranes. These changes were attributed to the accommodation of cleaved chains within the film, particularly evident in the binary mixture and to a lesser extent in the ternary and quaternary mixtures. The effect of oxidation on the film behavior was significant, affecting its surface activity, morphology, viscoelasticity and stability. Oxidation resulted in fluidization of binary and ternary blend films, disruption of the growth of CO-condensed phase domains and multilayer behavior, and alteration of the phase transition of GT. The presence of FFAs, which do not undergo oxidation, affected the line tension and increased the surface coverage of the condensed phase after oxidation. The effect of oxidation on film dilatational elasticity and viscosity was found to be dependent on film composition, highlighting the importance of using appropriate model membranes. The study highlighted that elevated ozone concentrations can have significant effects on TFLL model membranes, which are critical for the proper functioning of the physiological TFLL.

## 5. The Lung Surfactant Films (LS) and the Inhalation of Pollutants

Lung surfactant (LS) is a complex mixture of lipids (approximately 90% by weight) and proteins (approximately 10% by weight) [124,125]. The specific proteolipidic composition of LS plays a crucial role in controlling its structure, properties, and function. This process results in the formation of a surfactant layer on the surface of the alveolar epithelium,

which effectively reduces interfacial tension and aids in respiratory mechanics [126,127]. In addition, the LS film also acts as an important biological barrier to prevent inhaled particles from entering the mammalian bloodstream. Figure 5 shows the average composition of mammalian LS.

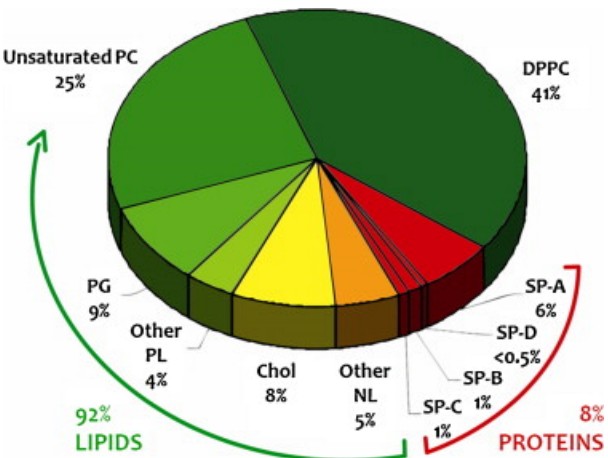

**Figure 5.** Average composition, as a percentage of the total weight, of mammalian lung surfactant (LS). The composition includes the following components: phospholipids (PL), phosphatidylcholines (PC), phosphatidylglycerol (PG), cholesterol (Chol), neutral lipids (NL), and surfactant proteins A/B/C/D (SP-A/B/C/D). Reprinted with permission from Ref. [125]. 2015, Elsevier.

The composition of mammalian lung surfactant (LS) involves an intricate interaction between different elements, including phospholipids, cholesterol, neutral lipids, and various surfactant proteins. Each element plays an essential role in the performance and resilience of LS and is central to the respiratory processes and defense mechanisms of the lung.

The lipid fraction of mammalian lung surfactant (LS) is composed of three major components: phospholipids, cholesterol, and neutral lipids, which account for approximately 79%, 8%, and 5% of the total LS weight, respectively. Among the phospholipids, the zwitterionic phosphatidylcholines (PC) play a crucial role, constituting a substantial part (about 60–70%) of the total weight of the LS. In addition to PC, anionic species such as phosphatidylglycerols (PG), phosphatidylinositols (PI), phosphatidylethanolamines (PE), and sphingomyelin are present at lower concentrations [128,129]. A more detailed analysis shows that 1,2-dipalmitoyl-sn-glycero-3-phosphatidylcholine (DPPC) is the major component of the lipid fraction, accounting for approximately 40–50% of the total LS weight [130]. Interestingly, the exact content of DPPC in LS appears to be inversely proportional to the rate of surface area change, which is linked to the respiratory rate during breathing [124]. In addition, other phosphatidylcholines commonly found in mammalian LS include 1-palmitoyl-2-oleyl-sn-glycerol-3-phosphatidylcholine, 1-palmitoyl-2-palmitoleyl-sn-glycero-3-phosphatidylcholine, and 1-palmitoyl-2-myristoyl-sn-glycero-3-phosphatidylcholine. The specific ratio of DPPC to other phosphatidylcholines correlates with respiratory rate and alveolar development [128,131–134].

Approximately 10% of the total weight of lung surfactant (LS) is composed of anionic phospholipids, including phosphatidylglycerols and specific phosphatidylinositols. It is noteworthy that these compounds lack myristoyl moieties and have a low concentration of dipalmitoyl phosphatidylglycerol [128,135]. The presence of anionic phosphatidylglycerols and phosphatidylinositols is important in facilitating interactions between the lipid fraction and the cationic surfactant proteins (SP-B and SP-C) [67].

Cholesterol is another critical component of the LS as it modulates the packing of lipids within the LS structures. Changes in cholesterol concentration can have a profound effect on the lateral arrangement and performance of lung surfactant (LS) [136]. In addition, neutral lipids, which include cholesterol esters, triglycerides, diglycerides, and free fatty

acids, play an important role in surfactant properties. The composition of the neutral lipid fraction has evolved over time to enhance surfactant activity. [137].

The lung surfactant (LS) protein fraction consists of four primary surfactant proteins: SP-A, SP-B, SP-C, and SP-D, listed in the order of their original discovery. In certain species, additional proteins such as SP-G and SP-H may also be present [138]. Of these, SP-A and SP-D are hydrophilic and play a critical role in lung defense mechanisms and the maintenance of surfactant balance [139]. However, the hydrophobic SP-B and SP-C control the interfacial properties of LS and influence its overall functionality [130,140]. SP-B (molecular weight ~8.7 kDa) is positively charged and α-helical, critical for LS adsorption at the liquid/gas interface and dynamic exchange with low surface activity compounds [141,142]. It is essential for LS storage and SP-B deficiency can be lethal [143]. SP-C (molecular weight ~3.7 kDa) is a small hydrophobic protein important for vesicle fragmentation and membrane curvature [144,145].

## 5.1. Interfacial Activity of Lung Surfactant

LS interfacial activity depends on phospholipid self-organization at the interface, LS composition, temperature, and lateral pressure [126,146]. Below the melting temperature, phospholipid lateral mobility is hindered by strong interfacial packing, while above the melting temperature, fluidity increases. The transition from ordered to disordered phases depends on the molecular properties of the phospholipids. In the case of the primary component of lung surfactant (LS), DPPC, this transition occurs at approximately 41 °C. In contrast, phospholipids with unsaturated acyl chains can undergo this transition at temperatures below 0 °C. Cholesterol plays a crucial role in the modulation of lipid packing. In interfacial films (monolayers), phase behavior is associated with changes in surface pressure. At low interfacial density of phospholipids, the behavior of the interface is reminiscent of a gas-like monolayer, while higher densities push the monolayer through higher lateral order phases, hindering lipid mobility, resulting in higher order phases such as the liquid expanded phase (LE) or the liquid condensed phase (LC), eventually reaching a 2D solid-like conformation for high Π values [147].

Lipid membranes have the ability to adopt various three-dimensional structures when immersed in aqueous environments, including both lamellar and non-lamellar configurations. The distinctive molecular properties of lipids, particularly their hydrophilic-lipophilic balance (HLB), play a critical role in the adsorption of lung surfactant (LS) at the liquid/gas interface and its reorganization during the respiratory process [126]. Figure 6 illustrates the different lipid structures that can manifest in LS under physiological conditions.

The lipid polymorphism in LS surfactant is critical for proper respiratory function and fulfills three conditions: efficient interfacial adsorption, low interfacial tension during compression without compromising stability, and effective redistribution during expansion. De novo secreted lung surfactant (LS) undergoes a continuous adsorption process at the gas/liquid interface, which is influenced by LS composition, lipid concentration and structure. The key step in achieving functional interfacial film formation (see Figure 6) is the transition from a bilayer to a monolayer configuration. This transition requires non-lamellar structures involving anionic lipids as well as SP-B and SP-C proteins [146,148–151]. SP-B plays a critical role, especially during the initial interaction of LS with the interface. It promotes membrane aggregation, fusion, and permeabilization, facilitating the rapid exchange of lipids between the interfacial film and the underlying fluid phase (reservoirs) [141,142,152–156].

LS adsorption and spreading at the gas/liquid interface results in an interfacial film with significantly reduced interfacial tension during expiration, minimizing work of breathing [67]. To further reduce the interfacial tension during the exhalation phase, the composition of the interfacial film can be modified [157] by excluding molecules with lower capabilities for reducing interfacial tension (unsaturated phospholipids, cholesterol and surfactant proteins) while enriching the interface with DPPC. This creates solid structures under high lateral pressures that increase lung compliance and stabilize alveolar volume,

preventing premature alveolar collapse during expiration [125,158,159]. The association of reservoirs of excluded lipids and proteins with the interface facilitates compositional remodeling during inspiration, adding new molecules to the LS film [39,156,160–164]. This structural interplay ensures an efficient minimization of interfacial tension, driven by desaturated lipids and supported by surface-active proteins (SP-B and SP-C).

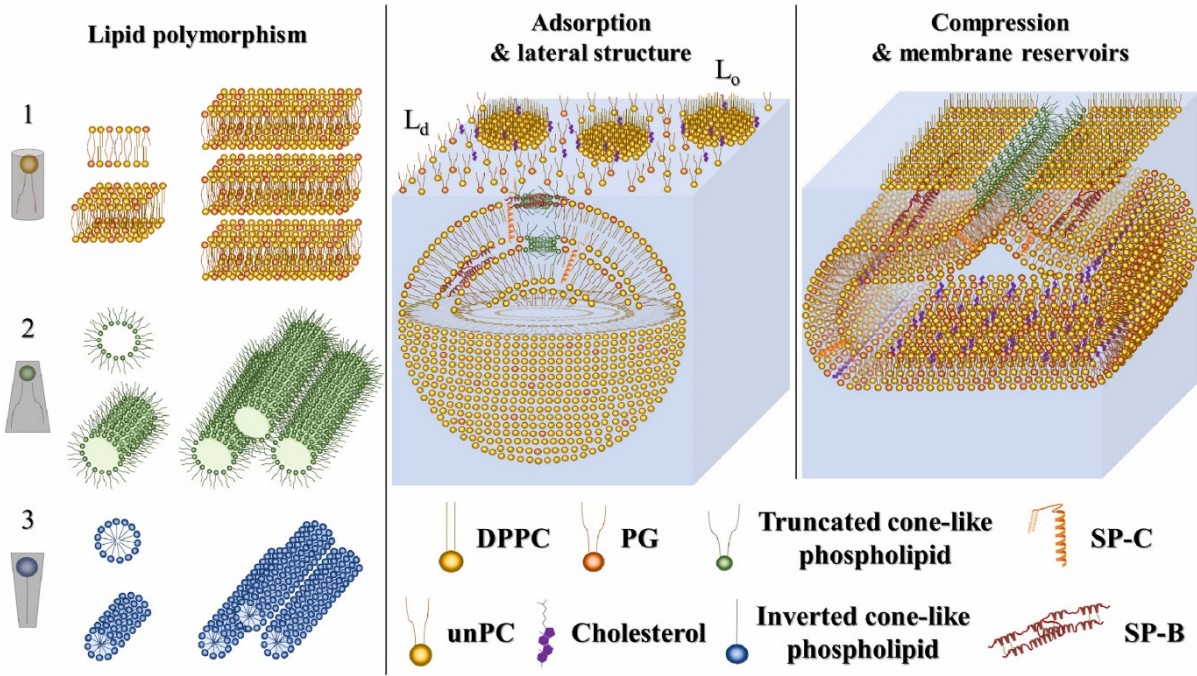

**Figure 6.** Different structures adopted by LS. Reprinted from Castillo-Sánchez et al. [126].

The lateral arrangement within the lung surfactant (LS) interfacial film resembles the coexistence of two distinct liquid phases, characterized by DPPC-enriched domains surrounded by a disordered lipid phase. These domains expand during exhalation, eventually forming a solid structure at maximum compression. During inhalation, this structure reverts due to rapid lipid replenishment from reservoirs, leading to an increase in interfacial tension and a transition to the disordered phase. This continuous adaptation of the LS films under the guidance of the surface proteins SP-B and SP-C, which link the interfacial film to the reservoirs, facilitating effortless breathing [126,127,165]. Figure 7 provides a visual representation of the processes occurring during the compression–expansion cycles within the alveolar cavity, illustrating the remodeling phenomena and their effect on interfacial tension.

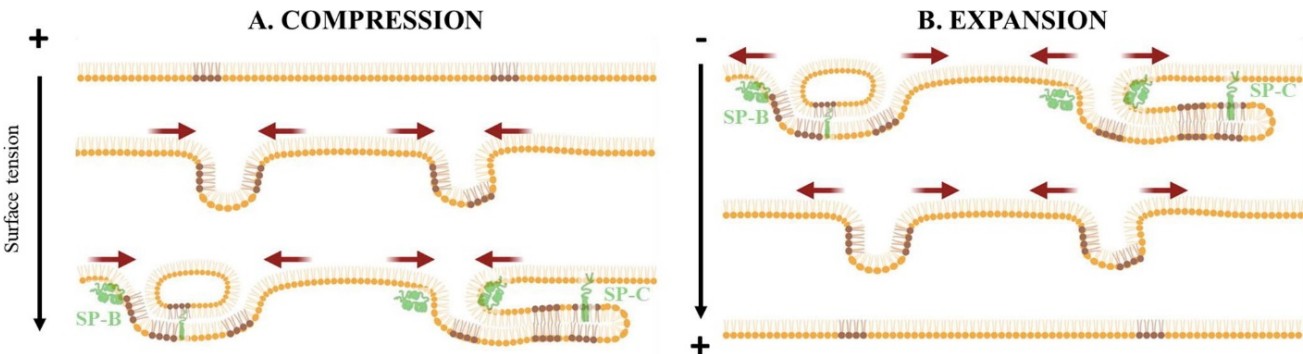

**Figure 7.** Illustration of diverse remodeling processes during compression-expansion cycles of the alveolar cavity. Reprinted with permission from Ref. [127]. 2021, Elsevier.

It is important to emphasize that the recycling of lipids and proteins that are excluded from the interfacial layer due to the actions of alveolar macrophages and Type II pneumocytes is critical for maintaining normal lung surfactant (LS) interfacial activity [166]. The central role in this recycling process is played by the SP-C protein, which contributes to the formation of small vesicles that are readily engulfed by cells [144]. In addition, SP-B stimulates the secretion of lamellar bodies in Type II pneumocytes [167], while SP-A plays a role in regulating the inhibition of LS secretion when their concentration in the alveoli is sufficient [168].

*5.2. Lung Surfactant Models for In Vitro Studies*

Various mixtures are used to study lung surfactant (LS) films, ranging from simple lipid monolayers to complex surfactant extracts from bronchoalveolar lavage fluid. In order to draw biophysically relevant conclusions, it is crucial to understand the physicochemical differences between the chosen model and the actual lung surfactant. An ideal surfactant model should replicate specific physical properties relevant to the physiological role of surfactant, such as the ability to reduce interfacial tension, undergo effective compositional restructuring during compression, and exhibit rapid reabsorption and redistribution during expansion [10,32,45,169].

DPPC fulfills the first requirement and is often used as a minimal model to study LS films. However, its inefficiency in reservoir formation and slow re-spreading limits its application in understanding LS performance [5,32,73,170–174]. To address this challenge, researchers employ more intricate models that involve combining DPPC with other lipids or fatty acids, such as palmitic acid, cholesterol, or DOPC (1,2-dioleoyl-sn-glycero-3-phosphocholine). These models provide valuable insights into the behavior of lung surfactant (LS) under physiological conditions. Specific lipids such as POPC (1-palmitoyl-2-oleoyl-sn-glycero-3-phosphocholine) or POPG (1-palmitoyl-2-oleoyl-sn-glycero-3-(phospho-rac-(1-glycerol))) play an important role in modulating the phase behavior of LS films [71,175–177].

The use of single lipids or their combinations as models for lung surfactant (LS) can be challenging in terms of biophysical interpretation. The absence of surface-active proteins complicates the control of molecular exchange between the interface and the adjacent aqueous subphase during compression-expansion cycles. Realistic models using natural extracts or laboratory mixtures that closely resemble mammalian LS compositions are essential to gain a comprehensive understanding of how particles affect LS films. Commercial lung surfactant (LS) formulations, commonly used in surfactant replacement therapy and the treatment of neonatal respiratory distress syndrome, have received increasing attention as models for understanding in vitro respiratory physiology. This includes the investigation of potential therapeutic effects against SARS-CoV-2. [67,160,162,178]. Table 2 provides a summary of some of the clinical pulmonary surfactants currently in use and their respective sources.

**Table 2.** Overview of common clinical lung surfactants, including type, origin, and producer. Adapted from Guzmán [2], with permission under Open access CC BY 4.0 license, https://creativecommons.org/licenses/by/4.0/ (accessed on 24 July 2023).

| Commercial Name | Origin | Producer |
|---|---|---|
| Infasurf® | Lavage of calf lung fluid | ONY Biotech Inc., Amherst, NY, USA |
| Curosurf® | Lavage of porcine lung fluid | Chiesi Farmaceutici S.p.A, Parma, Italy |
| Survanta® | Lavage of bovine lung fluid | AbbVie Inc., North Chicago, IL, USA |
| BLES® | Lavage of bovine lung fluid | BLES Biochemicals Inc., London, ON, Canada |
| Alveofact® | Lavage of bovine lung fluid | Lyomark Pharma, Oberhaching, Germany |
| Venticute® | Synthetic | Byk Gulden Pharmaceuticals, Konstanz, Germany |
| Surfaxin® | Synthetic | Discovery Laboratory Inc., Warrington, PA, USA |
| Exosurf® | Synthetic | GlaxoSmithKline, Brentford, UK |

The comparison of the behavior of interfacial films of DPPC and commercial LS formulations in the presence of colloidal particles at fluid interface demonstrated the importance of importance of using realistic LS models for obtaining physically sound conclusions [179]. DPPC films with particles undergo a rapid destabilization, leading to a premature collapse and the formation of 3D structures that are not easily readsorbed upon expansion [180,181]. However, commercial LS formulations containing other lipids and surface-active proteins SP-B and SP-C show minimized destabilization, allowing for proper remodeling of the interfacial layer [179,182–184]. A similar conclusion can be drawn from the analysis of the minimum interfacial tension achieved by LS models in the presence of particles. When particles are introduced into the interfacial layers of commercial lung surfactant (LS) formulations, there is a smaller increase in the minimum interfacial tension during compression than when particles are introduced into DPPC layers alone. This difference can be explained by the presence of surface-active proteins that assist in the interfacial remodeling process. These proteins play a key role in the formation of reservoirs and the kinetics of re-adsorption and re-distribution of expelled material during interfacial expansion [179,185].

The Tanaka mixture of lipids (DPPC, POPG, and palmitic acid In a weight ratio of 68:22:9) serves as a simple and robust model for studying the interfacial properties of lung surfactant (LS) [186]. The incorporation of POPG into DPPC monolayers plays a critical role in modulating the interfacial compressibility of the film, resulting in a reduction of the inherent stiffness observed in pure DPPC films. This enhanced flexibility is critical to the packaging of LS and its ability to reduce interfacial tension. In addition, POPG contributes significantly to the formation and stabilization of reservoirs. Conversely, palmitic acid is instrumental in promoting film fluidity at high interfacial tensions, facilitating molecular rearrangements within LS films. In addition, palmitic acid enhances interfacial packing, resulting in improved film stiffness at low interfacial tensions, effectively preventing premature collapse.

While numerous studies on LS models focus on single monolayers, a more accurate representation of LS complexity can be achieved by including other colloidal structures, such as bilayers or multilayers [8]. By adopting this approach, researchers can gain valuable insights into the behavior of LS and how endogenous aspects influence it, particularly when considering the significant role of bulk structures such as reservoirs. These structures play a key role in facilitating the exchange of material between the interface and the adjacent fluid phase during compression-expansion cycles [187]. Regarding the mechanical aspects of lung surfactant (LS) films, there remains a vast field for exploration. However, models that aim for biophysical relevance must take into account two primary factors. First, the temperature used should closely be to physiological conditions (37 °C). Second, specific hydrodynamics conditions inherent to the experiments that govern Marangoni flows within the interfacial film and between the interface and the adjacent fluid phase, should be carefully considered. [32,163,164,188–190].

*5.3. Impact of Pollutants in Lung Surfactant Films*

Air pollution is a major public health concern, linked to various cardiovascular and respiratory diseases. This is particularly important because, in recent years, there has been a significant increase in the release of pollutants into the atmosphere from combustion processes in industry, power plants, heating systems, vehicles, and nanotechnology production residues. This raises concerns about the potential adverse effects on human health [191]. Recent World Health Organization (WHO) statistics indicate that long-term exposure to air pollutants is a major contributor to cardiovascular disease and mortality, with about one-third of deaths from stroke, lung cancer, and heart disease being related to air pollution [192–194]. Consequently, contemporary society faces a pressing public health problem in the form of air pollution [195,196]. This requires a comprehensive assessment of the potential effects of pollutants on regular physiological processes, with a particular focus on inhaled pollutants [169,197,198].

The inhalation of pollutants has been linked to the development of pathological conditions and respiratory diseases, including but not limited to acute respiratory distress syndrome (ARDS), asthma, chronic obstructive pulmonary disease, fibrosis and lung cancer [199,200]. Therefore, careful investigation of the effects of particles on respiratory function is essential [201]. Among pollutants, fine particulate matter (PM$_{2.5}$, particles $\leq$ 2.5 µm in diameter) is considered one of the most important sources of respiratory disease [202–204]. Inhaled particles can travel through the respiratory tract and reach the alveoli where they come into contact with the lung surfactant (LS) film [205,206].

Several mechanisms are involved in the transport of inhaled particles through the respiratory tract, including Brownian diffusion, gravitational sedimentation, inertial impaction, and interception [207]. However, due to the complex nature of respiratory dynamics, the precise contribution of each process remains unclear. Inhaled aerosols typically exhibit significant polydispersity, which limits their ability to penetrate deep into the lung. Only a small fraction of particles smaller than 4 µm manage to reach the alveolar region. The tortuous path with 23 tubular bifurcations presents obstacles to particle movement from the initial inhalation through the nasal passages to the alveoli [169].

The biophysical consequences of inhaled substances typically begin when they enter the alveoli, leading to partial inhibition of lung surfactant (LS) function and affecting overall lung function [127]. Despite the low concentrations of particles that are typically deposited on the large alveolar surface (up to 100 m$^2$), the high surface-to-volume ratio of particles results in stronger effects than expected when considering only the deposited mass (acquired dose) [47,185,188,208–210]. Therefore, several parameters, including particle size, concentration, molecular structure, hydrophobicity, and charge, must be considered when assessing the safety of inhaled particles [35,38,207,211–217].

Particle deposition on the LS layer involves multiple events triggered by particle-LS interactions. These interactions can occur directly with specific LS film components at the interface or indirectly by competing with LS components for interfacial space [14,36,37,218,219]. Such interactions can inhibit LS functionality and critically affect normal respiratory function [32]. The modification of Marangoni flows within lung surfactant (LS) upon particle deposition plays a pivotal role, as these flows govern the compositional reconfiguration of the LS film during the respiratory cycle [189,190]. While alveolar compression, especially under conditions of high interfacial tension, could lead to collapse, the ability of the LS to reduce interfacial tension during exhalation prevents such collapse due to its high content of DPPC and other desaturated phospholipids, which form densely packed phases even at extremely low interfacial tensions. However, particle incorporation may interfere with the mechanical aspects of normal respiratory function [32,127,188].

Particle deposition on the lung surfactant (LS) film has the potential to hinder the formation of condensed phases during alveolar compression, limiting the reduction of interfacial tension and potentially causing premature alveolar collapse. For example, hydrophobic particles may become trapped in hydrophobic regions of the LS film, altering interactions and impeding the formation of dense phases [77,180]. This interference may prevent the interfacial tension from reaching low levels during compression, making it difficult to counteract the forces pulling the alveolar walls together [220]. While collapsed alveoli may reopen during inspiration, there is a reduction in tidal volume [127]. Particle incorporation alters the dynamics of LS film remodeling by changing the area of the hysteresis loop associated with the compression-expansion cycles. These changes result from altered boundary conditions governing mass transport between the interfacial film and the adjacent fluid. They result from either incomplete clearance of inhaled particles from the LS film or a shift in fluid phase composition due to particle expulsion during compression [60,189,190,221,222]. In addition, particles can affect LS functionality by adsorbing LS compounds onto their surfaces, forming an LS corona. This corona affects particle wettability and LS composition, which subsequently affects interfacial tension reduction and the remodeling process [223]. In particular, SP-A has a known affinity for binding to hydrophilic particles, facilitating the subsequent adsorption of other LS

components and aiding particle clearance from the lung [224]. However, the scenario is different for hydrophobic particles, as their interaction with hydrophobic components initiates corona formation [225]. Consequently, particles entering the alveolar cavity may become embedded in the LS membrane and undergo dynamic compression-expansion processes. This may eventually lead to their release from the LS into the alveolar spaces, allowing for interaction and potential internalization across the cellular barrier [226,227]. As a result, particle inhalation may have both short- and long-term adverse effects on the normal physiology of the respiratory process.

Langmuir Films for Evaluation of the Interaction of Particles with LS Film

To understand the potential adverse consequences associated with particle entrapment in LS layers, a thorough investigation of their effect on tensiometric properties, dynamic response of interfacial films, lateral organization and structure is imperative. In addition, evaluation of the distribution of particles between phases of different order is crucial, as this can significantly influence particle translocation through the lung fluid and clearance processes.

Once deposited on the lung surfactant (LS) layer, particles possess distinct physicochemical properties that determine the nature of their interactions with the LS and the resulting effects. Furthermore, it is imperative to control the number of particles interacting with the LS layer, since, in most cases, the adverse effects associated with inhaled particles are highly dependent on the particle dose [228–232].

It is crucial to emphasize that particles typically affect respiratory function at two different levels: (i) The biological functionality of lung surfactant (LS); and (ii) The metabolism of LS [169]. Consequently, understanding the effects of particles on the physicochemical properties of LS layers requires a thorough investigation of the particular properties of the particles under study [36,188].

The chemical nature of particles strongly influences their interaction with LS films. Unfortunately, it is difficult to systematize the effect of particle chemistry on LS performance [5,188]. However, general aspects of the influence of particle chemistry on LS films can be drawn. Silicon dioxide particles show a meaningful change in LS performance that is closely related to the number of SiOH groups present on their surface. The higher the surface density of silanol groups, the greater the likelihood of developing silicosis and other lung diseases associated with the inhalation of silicon dioxide particles [233].

The importance of particle chemistry in altering the properties of lung surfactant (LS) is illustrated by comparing the effects of carbon black and fumed silica on the interfacial properties of DPPC layers [36]. Despite their physical similarities, the different chemical properties of these particles dictate the equilibrium of interactions at the interface, thereby influencing the degree of interfacial perturbation and particle aggregation at the interface. Figure 8 provides a concise overview of how the different physicochemical properties of the particles affect the LS function.

Related to the chemical nature of particles are surface charge and hydrophilic–lipophilic balance (wettability), which affect the ability of particles to interact with biological structures. In fact, these physicochemical properties play a critical role in the incorporation of particles into biological interfaces, affecting interactions with biomolecules and influencing the metabolic pathways and biophysical function of LS [188,227,234]. For instance, charged polystyrene particles were found to induce a greater inflammatory response in the lung compared to neutral particles [235], indicating the prominent role of electrostatic interactions in modifying respiratory physiology [181,228,236].

The interaction between negatively charged hydrophilic silica particles and DPPC monolayers is strongly influenced by electrostatic forces, resulting in changes in the behavior of lipid molecules at the interface and hindering the formation of ordered phases. Conversely, hydrophilic silica particles are efficiently displaced from the interface at high surface pressures, suggesting the formation of a lipid corona surrounding the particle surface [221]. Similar results have been observed with negatively charged polystyrene

carboxylate particles interacting with DPPC films [209]. The inhibitory effect on lung surfactant (LS) function due to interactions with charged particles has also been demonstrated by molecular dynamic simulations [237]. It is important to note that the specific charge of the particles does not significantly affect their interaction with monolayers formed by zwitterionic lipids such as DPPC. However, the effect of cationic and anionic particles on the actual LS may be more complicated due to the presence of lipids and proteins with varying positively and negatively charged regions. Behyan et al. [175] independently demonstrated the different effects of charged particles on the interfacial properties of lipid blends and protein-containing LS formulations. Their investigation of the interaction of positively and negatively charged silica particles with different LS models, including a DPPC/POPG mixture and calf lung extract (Infasurf®), showed that the behavior of Infasurf® was influenced solely by cationic particles, whereas the DPPC/POPG mixture was affected by both positively and negatively charged particles.

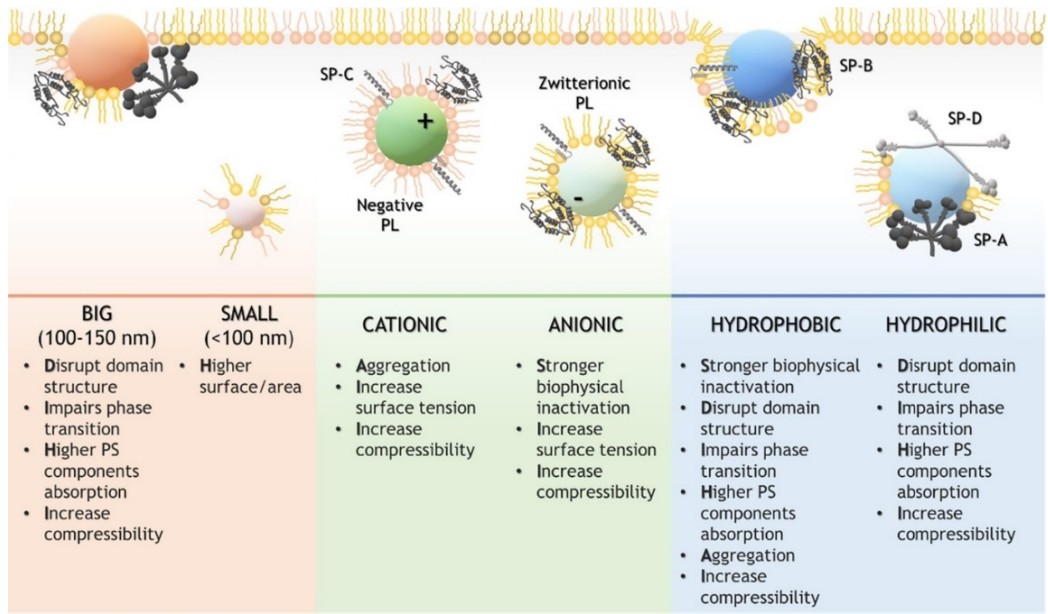

**Figure 8.** General view of the effect exerted by the physicochemical properties of particles when enter in contact with LS films. Reprinted with permission from Ref. [169]. 2019, Elsevier.

Negatively charged polylactide particles strongly inhibit the behavior of bovine LS extract (Curosurf®) due to their association with the SP-B protein. In addition, the interaction of these particles with the SP-C protein also contributes to their inhibitory effect. Conversely, positively charged particles have a reduced inhibitory effect, highlighting the importance of their charge nature in the inhibition of LS behavior [157,238,239]. This is consistent with the different results observed in the interaction of positively and negatively charged alumina, silica, and latex nanoparticles with LS layers. Negatively charged particles have a limited effect on LS function, while positively charged particles lead to the formation of aggregates with LS vesicles. The intensity of this interaction increases as the charge density of the particles increases [240].

The wettability of particles is another important parameter influencing their interaction with LS films [181]. Hydrophobic particles incorporated into DPPC layers modify the lateral packing of molecules at the interface and reduce the film rigidity, while hydrophilic particles undergo effective clearance upon compression [38,180]. Hydrophobic particles show enhanced retention within alveolar lining films due to strong van der Waals interactions with lipid molecules [210,241]. In general, the incorporation hydrophobic particles into DPPC films results in more expanded states due to an excluded area effect induced by particles, while hydrophilic particles induce more compressed states and reduce the average distance between lipid molecules [37,38,180,242]. However, when commercial LS

formulations are used as models, the isotherms appear to be shifted to more compressed states regardless of particle wetting properties, leading to a strong inhibition of LS performance characterized by a worsening of interfacial tension reduction and compositional remodeling [239,241]. Valle et al. show that the penetration of hydrophobic particles causes a contraction of the available area for LS molecules, resulting in a shift of the isotherm to more compressed states, which significantly affects surface tension reduction. Moreover, the incorporation of particles disrupts the lateral packing of molecules at the interface, hindering the formation of domains with ordered phases and disturbing the monolayer-to-multilayer transition, as shown by AFM micrographs of Langmuir-Blodgett films at different surface pressures. In addition, hydrophobic particles tend to aggregate within LS layers, demonstrating their key role in particle retention and translocation in the pulmonary fluid. Particle incorporation also increases the area of the hysteresis loop during the compression-expansion cycles of LS layers, which is expected to critically affect the normal function of LS [241].

The interaction of hydrophilic and hydrophobic particles with LS films follows different pathways. Hydrophilic particles penetrate LS layers directly, whereas hydrophobic particles require wrapping by LS components to be incorporated into LS films [243,244]. This difference influences the inhibition induced by hydrophilic and hydrophobic particles, with hydrophilic particles showing faster inhibition compared to hydrophobic ones [239,241]. A recent study by Beck–Broichsitter et al. [245] revealed that the inhibition of LS activity upon particle deposition is closely related to the ability of the particles to sequester surface-active proteins. Consequently, shielding the particles to prevent the formation of an LS corona reduces their harmful effects and allows their use as carriers for inhalable drugs.

Particle size also plays a critical role in particle-cell interaction, affecting particle uptake, cytotoxicity, and inflammatory response [246,247]. Consequently, it is expected to influence the interaction between inhaled pollutants and the LS film. The impact of particle size on LS films depends on the specific particle chemistry. In some cases, particle size is the main factor governing the interaction, while in others it becomes independent of size and relies on the nature of the particle [76,248].

Dwivedi et al. [210] studied the interaction of hydrophobic poly(organosiloxane) particles of different sizes (12 and 136 nm) with two LS models. Smaller particles showed minimal modification of the DPPC layers, while larger particles induced significant changes in the phase behavior of the DPPC film. When DPPG and SP-C were incorporated with DPPC in the LS model film, smaller particles caused less modification, but larger particles (136 nm) strongly altered the interfacial phase behavior of the LS model, particularly at high surface pressures. In addition, the larger particles led to a fluidization of the LS film, which reduced its elasticity. This may be explained by the disruption of the lateral packing of the LS film that occurs when particles are incorporated, causing changes in the cohesion between the molecules in the monolayer. This complexity arises from various interactions, such as steric hindrance, excluded area effects, and other factors [181]. As a result, particle size and concentration significantly influence the performance of the LS film [38,78,180,211].

Orsi et al. [76] observed that the incorporation of particles (size range: 9–60 nm) into DPPC films resulted in an interfacial organization similar to Pickering emulsions, leading to hindered domain growth. However, Ku et al. [249] found that the incorporation of gelatin particles into DPPC films was strongly dependent on the specific dimensions of the particles, with larger particles showing the strongest interaction and altering the interfacial behavior of the lipid. The impact of particle size on the physicochemical properties of LS models is unclear and controversial. Kodama et al. [250] found that only very small particles (around 20 nm) significantly modify the interfacial phase behavior of the Survanta® LS model. The complex interplay of factors, including particle and LS model type, as well as specific interfacial behavior, leads to varied scenarios for particle-LS interactions.

Particle shape, especially shape anisotropy, is also crucial for tuning the physicochemical properties of colloidal particles and their self-organization at fluid interfaces [231].

Therefore, it is expected to play a significant role in modifying the LS performance upon particle incorporation [251]. For instance, when comparing the impact of three types of particles on DPPC layers—two surface-inactive anisotropic clays (plate-like bentonite and halloysite nanotubes) and spherical silicon dioxide particles—it was found that increasing particle anisotropy facilitates particle clearance upon compression [242]. However, systematic experimental studies evaluating the influence of particle anisotropy on LS layers are currently lacking. Nevertheless, molecular dynamics simulations have shown that the length-to-diameter aspect ratio of particles plays a critical role in controlling their penetration and perturbation of the LS function [252].

The effect of particle anisotropy on LS layers was further explored by Kondej and Sosnowski [57], who investigated how carbon particles of different geometries (nanotubes and nanohorns) modify the performance of LS layers. They found that increasing the surface area of the particles leads to stronger frustration in LS behavior. Importantly, much of the effect of particle anisotropy on LS behavior can be attributed to the influence of capillary forces [253].

In addition to the properties discussed above, other physical parameters such as total surface area or specific surface area also influence the behavior of LS layers upon particle incorporation. This influence is due to both particle aggregation within LS layers and specific particle-LS interactions [250], which is consistent with in vivo observations of particle–LS interactions and deposition along the respiratory tract.

## 6. Challenges to Address in the Study of Lipid-Based Films at Fluid Interfaces

In recent years, the study of lipid-based films at fluid interfaces has garnered significant attention due to their pivotal role in various applications ranging from drug delivery and food science to cosmetics and biomaterials. However, this research landscape is not devoid of challenges, and as we delve deeper into understanding these interfaces, several complexities emerge that require further investigation.

One of the major challenges in the study of lipid-based films at fluid interfaces is to understand the dynamic interplay of lipids at fluid interfaces. These interfaces are characterized by complex molecular dynamics involving potential phase transitions, lateral diffusion, and conformational changes. Capturing these dynamic behaviors requires advanced experimental techniques and theoretical models. Future research efforts should focus on unraveling the real-time behavior of lipid molecules at fluid interfaces, potentially using state-of-the-art spectroscopic methods and molecular simulations to gain insight into their transient structures and interactions. The characterization of lipid-based films at fluid interfaces is a significant challenge. Traditional analytical techniques struggle with the limited accessibility of these interfaces. While methods such as Langmuir-Blodgett troughs and Brewster angle microscopy provide valuable insights, they may not be sufficient to capture the complexity of real-world applications. Newer techniques, such as synchrotron-based x-ray and neutron scattering, hold great promise for elucidating molecular arrangements and interactions at fluid interfaces. On the other hand, environmental factors exert substantial influence on fluid interfaces. Temperature, pH, and the presence of ions can all significantly affect the stability and behavior of lipid films. Understanding these effects is pivotal for applications across fields. Future research should delve into the interplay between these factors and the interfacial behavior of lipids, shedding light on their implications for drug delivery, food encapsulation, and other pertinent domains.

Lipid-based films have applications in biomimetic systems and biological interfaces. However, capturing the intricate complexity of natural lipid membranes and understanding how synthetic lipid films mimic these systems is a significant challenge. Achieving this will require interdisciplinary collaborations bridging materials science, biology, and medicine. Future research directions should aim to establish links between synthetic lipid films and biological counterparts, providing more accurate models for studying cell membranes and their interactions. In addition, real-world applications often involve lipid-based films in combination with other components such as proteins, polymers, and nanoparticles. Study-

ing the behavior of such multicomponent systems at fluid interfaces is crucial for the design of functional materials. Research should focus on understanding the synergistic or competitive interactions between lipids and other components to facilitate tailored applications in drug delivery, nanomedicine and beyond. Translating fundamental studies into scalable and engineering applications presents its own set of challenges. While laboratory-scale studies provide valuable insights, translating these findings into practical applications requires addressing issues related to stability, reproducibility, and manufacturing processes. Future research should involve collaborations between scientists, engineers, and industry partners to develop robust processes for producing lipid-based films at fluid interfaces on a larger scale.

Finally, advancements in computational modeling can offer a promising avenue to predict the behavior of lipid-based films at fluid interfaces. The integration of molecular simulations and machine learning techniques holds the potential to provide insights into interfacial properties, stability, and interactions. Developing predictive tools can significantly accelerate the design of lipid-based systems with tailored functionalities, opening up new horizons for applications across disciplines.

The study of lipid-based films at fluid interfaces holds immense promise for diverse applications. Overcoming challenges related to interfacial dynamics, characterization techniques, environmental influences, and complex multicomponent systems requires collective effort from researchers across disciplines. The future of this field lies in the convergence of innovative experimental techniques, advanced computational modeling, and collaborative approaches that bridge the gap between fundamental science and practical applications. As these challenges and opportunities are navigated, the potential to revolutionize drug delivery, food science, cosmetics, and biomaterials is both exhilarating and limitless.

## 7. Concluding Remarks

This review offers a valuable perspective into the applicability of fluid films as model systems for understanding different physicochemical aspects of different lipid-based layers with significant biophysical relevance in living organisms. The focus was primarily on the tear film lipid layer and the lung surfactant, which are complex mixtures of lipids and proteins that interact at liquid/vapor interfaces to perform their vital biological functions. We have presented selected examples that illustrate experimental instruments, theoretical methodologies, and model investigations that contribute to our understanding of unique and intricate scenarios that arise within biological systems.

The complex balance between structure and composition is critical for the proper functioning of these biological layers. Perturbations in the lateral organization of the tear film lipid layer or the surfactant film of the lung, caused by both endogenous and exogenous factors, can lead to various pathogenic conditions that disrupt the normal physiological role of these lipid-based films. The review has shed light on essential aspects related to these biological systems and their potential impact on the emergence of pathogenic states, deepening our understanding of the effect of changes in composition and dynamics on their overall physiological function.

While it is acknowledged that in vitro evaluations using colloidal and interfacial approaches may not fully replicate true biophysical situations, they still provide fundamental insights into relevant physicochemical aspects. In conclusion, the utilization of fluid films as models for lipid-based layers serves as a crucial element in comprehending the biophysical functionalities of these systems and in revealing potential pathological conditions stemming from alterations within them. However, certain physicochemical and mechanical aspects of existing models require further refinement, which calls for a combination of physicochemical models with complementary biophysical and medical studies to address these open questions and advance our knowledge in this field.

**Author Contributions:** The three authors have contributed equally to the preparation and writing of this work. All authors have read and agreed to the published version of the manuscript.

**Funding:** E.G. was funded by MICINN (Spain) under grant PID2019-106557GB-C21, and by the E.U. in the framework of the European Innovative Training Network–Marie Sklodowska–Curie Action NanoPaInt (grant agreement 955612). A.M. was funded from MICINN (Spain) under grants PID2021-129054NA-I00 and the IKUR Strategy under the collaboration agreement between Ikerbasque Foundation and Materials Physics Center on behalf of the Department of Education of the Basque Government. P.G.A. was funded by the Alexander von Humboldt Foundation.

**Institutional Review Board Statement:** Not applicable.

**Informed Consent Statement:** Not applicable.

**Data Availability Statement:** Not applicable. In the preparation of this work, no new data were produced.

**Conflicts of Interest:** The authors declare no conflict of interest. The funders had no role in the design of the study; in the collection, analyses, or interpretation of data; in the writing of the manuscript; or in the decision to publish the results.

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
