# Peer review of "Fluid Interfaces as Models for the Study of Lipid-Based Films with Biophysical Relevance"

_coatings, doi:10.3390/coatings13091560_

Round 1

Reviewer 1 Report

The manuscript “Fluid interfaces as models for the study of lipid-based films with biophysical relevance” by Pablo G. Argudo, Armando Maestro and Eduardo Guzmán is a review of recent articles on the lipid layers at the liquid – gas interface, which are used as models of the layers of lung surfactants and tear films. Besides, the authors consider in details the interactions of the lipid monolayers with different nano- and microparticles. The relevance of this subject becomes obvious if we take into account a recent pandemic and the environment degradation. At the same time, note that although the both tear films layer and layers of the lung surfactant are very complex systems consisting of many different components, the most of recent studies can be reduced to the determination of some changes in biological systems as a result of the change of the composition without a serious analysis of the physico-chemical causes of the observed behavior. This approach inevitably is reflected in the review under consideration and the role of various components in bioprocesses is not elucidated to the sufficient extent. This task is very complex but some generalizations presumably can be achieved. This remark relates first of all to the section 3.1.2. Therefore the recommendation is to finish this section with some general conclusions form published articles.   

The manuscript by P.G. Argudo et al. is recommended for publication in Coatings after this correction.

Some misprints:

The symbol DPPC is explained four times in the manuscript, ls. 462, 423, 530 and 554.

Both tables in the manuscript are designated as Table 1.

L. 399. Please exclude a repetition.

Author Response

The manuscript “Fluid interfaces as models for the study of lipid-based films with biophysical relevance” by Pablo G. Argudo, Armando Maestro and Eduardo Guzmán is a review of recent articles on the lipid layers at the liquid – gas interface, which are used as models of the layers of lung surfactants and tear films. Besides, the authors consider in details the interactions of the lipid monolayers with different nano- and microparticles. The relevance of this subject becomes obvious if we take into account a recent pandemic and the environment degradation. At the same time, note that although the both tear films layer and layers of the lung surfactant are very complex systems consisting of many different components, the most of recent studies can be reduced to the determination of some changes in biological systems as a result of the change of the composition without a serious analysis of the physico-chemical causes of the observed behavior. This approach inevitably is reflected in the review under consideration and the role of various components in bioprocesses is not elucidated to the sufficient extent. This task is very complex but some generalizations presumably can be achieved. This remark relates first of all to the section 3.1.2. Therefore the recommendation is to finish this section with some general conclusions form published articles.  

Following the reviewer recommendation, we have extended the section 3.1.2 with some specific details about the role of the different components of the TFLL.

The manuscript by P.G. Argudo et al. is recommended for publication in Coatings after this correction.

Some misprints:

The symbol DPPC is explained four times in the manuscript, ls. 462, 423, 530 and 554.

We have avoided the repetition of the explanation of DPPC meaning.

Both tables in the manuscript are designated as Table 1.

We have corrected the designation of the Tables.

  1. 399. Please exclude a repetition.

We have removed the repetition.

We thank to the reviewer for the comments, they have been very useful for improving the quality of our manuscript.

Reviewer 2 Report

Dear author,

The review article entitled: Fluid interfaces as models for the study of lipid-based films with biophysical relevance, has high interesting of many readers. I suggest the authors to support the review by addressing the following suggestions:

1-     Applications of Lipid-Based Films and Fluid Interfaces:

-        Present practical applications of lipid films in areas like drug delivery, food science, and cosmetics.

-        Discuss how insights from fluid interface studies can be applied to improve these applications.

-        Highlight recent advancements and breakthroughs in utilizing lipid films and fluid interfaces.

2-      Challenges and Future Directions:

-        Address the challenges in studying lipid-based films at fluid interfaces, such as complexity and variability.

-        Suggest potential strategies to overcome these challenges.

-        Outline future research directions, emphasizing the integration of advanced techniques and interdisciplinary approaches.

One more thing, the author can use the following references:

-        "Synthesis of a new hydrophobic coating film from stearic acid of buffalo fat." Scientific Reports 12, no. 1 (2022): 18465.

Author Response

Dear author,

The review article entitled: Fluid interfaces as models for the study of lipid-based films with biophysical relevance, has high interesting of many readers. I suggest the authors to support the review by addressing the following suggestions:

1-Applications of Lipid-Based Films and Fluid Interfaces:

-Present practical applications of lipid films in areas like drug delivery, food science, and cosmetics.

-Discuss how insights from fluid interface studies can be applied to improve these applications.

-Highlight recent advancements and breakthroughs in utilizing lipid films and fluid interfaces.

Even though, we consider that the topic is a little far from the scope of the review, we have included a new section discussing the aspects proposed by the reviewer.

2-Challenges and Future Directions:

-Address the challenges in studying lipid-based films at fluid interfaces, such as complexity and variability.

-Suggest potential strategies to overcome these challenges.

-Outline future research directions, emphasizing the integration of advanced techniques and interdisciplinary approaches.

Following the reviewer recommendation, we have included a section devoted of the challenges on the study of lipid films at fluid interfaces.

One more thing, the author can use the following references:

-"Synthesis of a new hydrophobic coating film from stearic acid of buffalo fat." Scientific Reports 12, no. 1 (2022): 18465.

We have included the reference recommended by the reviewer.

We thank to the reviewer for the comments, they have been very useful for improving the quality of our manuscript.

Round 2

Reviewer 2 Report

Dear Author, 

Thank you for your consideration

Regards